# CSAPR: Complex-Scenario-Aware Prompt Refinement for Text-to-Video Generation

## Abstract

Recent years have witnessed rapid progress of diffusion models, which significantly advance the development of Text-to-Video (T2V) generation. Compared to Text-to-Image (T2I) generation, T2V models encounter additional challenges, including temporal consistency, motion coherence, and adherence to physical constraints across frames. To address these challenges, we propose a novel two-stage framework, i.e., Complex-Scenario-Aware Prompt Refinement (CSAPR), to improve prompt the quality for T2V generation. CSAPR consists of two stages, i.e., prompt refinement and prompt verification. In the prompt refinement stage, CSAPR classifies user prompts into one of eight representative categories and applies targeted rewriting strategies guided by predefined meta prompts. In the prompt verification stage, CSAPR aligns semantic atoms from the original prompt with decomposed chunks of the refined prompt, ensuring that the refined prompt faithfully preserves the intended semantics while avoiding inconsistencies. Extensive experiments on three benchmarks, i.e., VBench, EvalCrafter, and T2V-CompBench, demonstrate that CSAPR significantly improves alignment with user intent and video generation quality in complex scenarios (up to 1.40% in terms of average score).

## 1 Introduction

The rapid advancement of diffusion models (Peebles & Xie, 2023; Rombach et al., 2022) has revolutionized Artificial Intelligence Generated Content (AIGC), with applications ranging from image and video generation to 3D content creation (Guo et al., 2025; Huang et al., 2025b; Lin et al., 2025; Zhang et al., 2025a), speech and audio synthesis (Luo et al., 2023; Liu et al., 2024b; Oh et al., 2024), and controllable editing (Lee et al., 2025; He et al., 2025; Wang et al., 2025d), driving new opportunities in entertainment, education, design, and human–computer interaction. Amid these advances, Text-to-Video (T2V) generation has emerged as a particularly compelling direction. Compared to Text-to-Image (T2I) generation, T2V demands visually realistic frames together with temporal coherence, motion dynamics, and adherence to physical and causal constraints, thereby posing a substantially more challenging yet impactful research problem.

Recent works (Hao et al., 2023; Zhan et al., 2024b;a) have demonstrated that the prompts optimized by Large Language Models (LLMs) yield high-quality and user-aligned outputs when fed into diffusion models. This improvement arises while user-provided prompts tend to be concise and may omit the detailed descriptions favored by AIGC models. Specifically, high-quality prompts are expected to provide precise character and scene descriptions, follow specific expression patterns and incorporate domain-specific terminology for stylistic expression (Parsons, 2022; Witteveen & Andrews, 2022; Brade et al., 2023; Zhan et al., 2024a).

However, existing prompt refinement approaches are generally designed for T2I tasks, which typically involve employing retrieval-augmented generation to expand prompts (Sun et al., 2024), tailoring prompts based on user preferences (Zhan et al., 2024b;a), and enhancing prompts through entity-specific descriptions (Ozaki et al., 2025). While these strategies are often sufficient for producing a single coherent image, they fall short when applied to T2V generation, which introduces substantially greater complexity. In video generation tasks, models are required to simultaneously guarantee temporal consistency, ensure motion coherence, capture causal dependencies, and comply with physical laws across frames. Although RAPO (Gao et al., 2025) attempts to enrich textual

descriptions by constructing a relation graph to retrieve terms relevant to user prompts, it primarily focuses on inter-object relations while overlooking challenges distinctive to video generation, such as abstract descriptions or temporal consistency. More importantly, RAPO relies excessively on relevance-based prompt retrieval and augmentation, which cannot provide a deeper understanding of the motivations behind prompt rewriting or explicit guidance for refinement in complex scenarios.

To address these challenges, we summarize seven representative complex scenarios that pose significant difficulties for diffusion models in generating videos that align with user expectations. Since each scenario introduces unique challenges for T2V generation, distinct strategies are required to address them systematically. Motivated by this insight, we propose a novel two-stage framework, i.e., Complex-Scenario-Aware Prompt Refinement (CSAPR).

CSAPR is designed to enable the prompt refinement model to understand the underlying reasons why a generation request may be difficult for a T2V model to fulfill, and to employ targeted rewriting strategies accordingly. To achieve this, CSAPR adopts a two-stage framework that includes prompt refinement and prompt verification. In the refinement stage, CSAPR classifies the user input into one of the seven challenging scenarios or identifies it as a non-challenging case. According to the classification result, a pre-defined rewriting guideline is selected and employed as a meta-prompt to guide the rewriting process. In the verification stage, CSAPR extracts the original prompt into atoms and decomposes the rewritten prompt into multiple chunks. A semantic alignment process is then performed to verify whether the rewritten chunks adequately preserve and reflect the meaning of the original atoms. If verification is successful, the refined prompt is delivered to the T2V model. Otherwise, the rewriting model is provided explicit feedback about missing or conflicting semantic elements and instructed to regenerate the refined prompt. The major contributions are summarized as follows:

- We summarize seven challenging scenarios in T2V generation tasks and propose targeted improvement strategies for each scenario. To our knowledge, this is the first work that investigates prompt refinement for video generation in complex scenarios.

- Our prompt rewriting strategy is grounded in understanding the challenges faced by T2V models and is explicitly guided by well-defined objectives throughout the rewriting process.

- We additionally design a prompt verification method to ensure that the rewritten prompts neither omits any user-specified elements nor introduces conflicts with the user prompt.

- Extensive evaluations on VBench, EvalCrafter, and T2V-CompBench demonstrate the effectiveness of our prompt refinement approach for T2V generation tasks.

## 2 RELATED WORK

In this section, we present the existing works on T2V generation and prompt refinement approaches.

### 2.1 TEXT-TO-VIDEO GENERATION

With the rapid progress of diffusion transformers and large-scale generative models (Rombach et al., 2022; Peebles & Xie, 2023), T2V generation (Singer et al., 2023; Chen et al., 2024a) has emerged as a pivotal task in content creation. Building on this trend, recent research has advanced T2V models along multiple directions. Architectural innovations have introduced scalable designs, such as expert transformers (Yang et al., 2025) and linear-complexity attention modules (Wang et al., 2025b), which improve both efficiency and model capacity. Training-free and plug-and-play inference approaches further enhance motion dynamics and spatial fidelity without additional training by exploiting cross-model integration and attention map analysis (Bu et al., 2025; Zhang et al., 2025b; Jagpal et al., 2025). Precise control over entity appearance and interactions has also been achieved through structured captions, instance-aware modeling, and LoRA-based customization (Feng et al., 2025; Fan et al., 2025; Huang et al., 2025a). Beyond visual quality, LLM-guided reasoning and external knowledge retrieval are employed to improve adherence to physical laws and factual accuracy (Xue et al., 2025; Yuan et al., 2025). Video editing frameworks exploit spatial-temporal guidance and attention control to extend T2V models for precise video modification (Wang et al., 2025e). Although these methods have made significant progress in various specific directions, they focus on common scenarios, while video generation for complex scenarios remains a challenging task. For instance,

reasoning about complex multi-entity interactions and maintaining coherent event-level narratives are still open problems, especially in scenes requiring abstract semantics or long-range temporal dependencies.

## 2.2 PROMPT REFINEMENT

Prompt refinement aims to automatically optimize user-provided natural language prompts into enhanced formulations that better align with the preferences of diffusion models. Early studies primarily infer user preferences or rewriting capabilities to guide prompt refinement. Prompt Refinement with Image Pivot (PRIP) (Zhan et al., 2024a) encodes visual preferences from linguistic prompts into latent representations, which are then decoded into refined text prompt. Capability-Aware Prompt Reformulation (CAPR) (Zhan et al., 2024a) dynamically adjusts rewriting strategies based on user capability and introduces configurable features for fine-grained control. Although effective, these approaches explicitly rely on user click logs, system-generation logs and other interaction data, which assume large-scale user engagement with text-to-image or text-to-video models and require explicit user consent for data collection. More recent studies incorporate retrieval-augmented generation (RAG) techniques to enhance prompts by retrieving semantically relevant terms and enriching the original descriptions. These methods either construct a dedicated prompt repository as an external knowledge base (Sun et al., 2024) or build a relation graph from training data (Gao et al., 2025). At inference time, they retrieve the most relevant exemplar prompts and integrate them with user input to compose a coherent, optimized reformulation. However, these RAG-based methods merely find high-similarity descriptions to enrich original prompts, without reasoning about the causes for rewriting or specifying explicit reformulation goals, limiting their ability to address complex generation scenarios.

## 3 COMPLEX-SCENARIO-AWARE PROMPT REFINEMENT

In this section, we first present the background and motivation. Then, we explain the scenario classification-based prompt refinement. Afterward, we propose our four-stage prompt verification.

### 3.1 BACKGROUND AND MOTIVATION

In real-world scenarios, T2V models may receive highly challenging generation requests, such as synthesizing abstract concepts or producing videos with intricate inter-object relationships. Unlike text-to-image tasks, T2V generation further entails challenges of maintaining temporal coherence and ensuring logical consistency across frames. For example, a prompt describing "a flower blooming" requires modeling fine-grained morphological changes over time, while "multi-person interactions" demands precise control over dynamic relationships between characters. Such scenarios are highly susceptible to bring up prompt ambiguity, leading to unexpected generated videos, which cannot precisely meet the original prompt intention. With our excellent rewriting strategy, prompt refinement transforms abstract or complex user intents into explicit structured descriptions that specify action sequences, spatial relationships, and physical constraints. As shown in Figure 1, the proposed approach CSAPR consists of two complementary stages, i.e., prompt refinement and prompt verification. CSAPR aims to unlock the full potential of T2V generation systems through precise scene descriptions and detailed depictions of event dynamics.

### 3.2 REFINEMENT STAGE: SCENARIO CLASSIFICATION-BASED PROMPT REFINEMENT

In this section, we propose our scenario classification-based prompt refinement, including scenario summarization, refinement strategies for each scenario, and scenario classification.

#### 3.2.1 SCENARIO SUMMARIZATION

In order to systematically identify representative challenges in T2V generation, we conduct a comprehensive review of recent benchmarks, surveys, and task-specific studies (Furuta et al., 2024; Liao et al., 2024; Mago et al., 2025; Sun et al., 2025; Wang et al., 2025c). In addition, we carry out an comprehensive examination of existing open source T2V models, e.g., Wan (Wang et al., 2025a), OpenSora (Zheng et al., 2024), CogVideo (Hong et al., 2022), and proprietary T2V systems, e.g.,

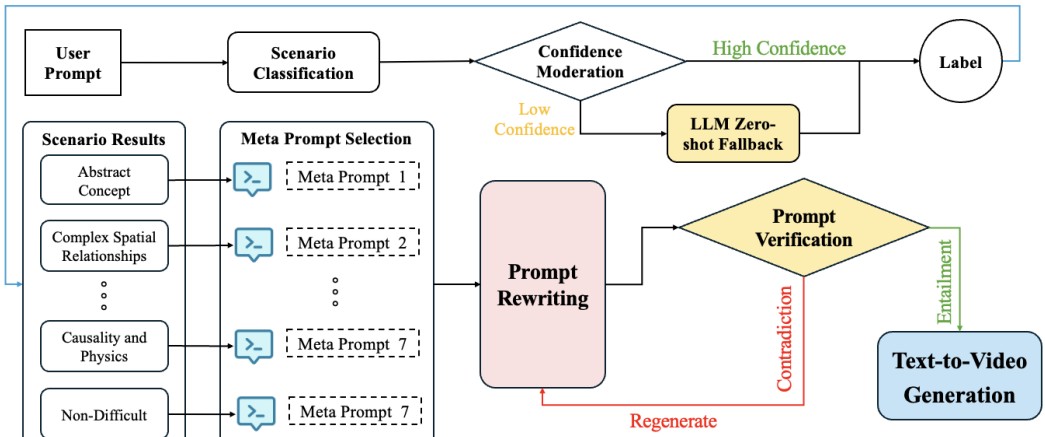

Figure 1: **Pipeline of Complex-Scenario-Aware Prompt Refinement (CSAPR).** CSAPR comprises two primary stages: prompt refinement and prompt verification. In the refinement stage, the input prompt is rewritten using a meta-prompt determined by its scenario classification. In the verification stage, the refined prompt is examined to ensure that it preserves the complete semantic content of the original prompt.

Veo3 (Google DeepMind, 2025), OpenAI Sora (OpenAI, 2024), Runway Gen-2 (Jennings & Runway, 2023).

From the review and the examination, we summarize seven representative categories of scenarios that are widely acknowledged as challenging for current T2V models. Abstract descriptions rely on metaphors or symbolism that require translation into concrete, visually interpretable content. Complex spatial relationships involve precise positions and orientations among multiple entities, where models often misrepresent layouts or occlusions. Multi-element scenes contain numerous characters, objects, and actions, demanding structured descriptions to avoid omissions or disorder. Fine-grained details emphasize subtle attributes such as facial expressions or textures, which models frequently blur or distort. Temporal consistency requires coherent progression of actions across frames, yet T2V models generally produce discontinuities or sudden jumps. Stylistic hybrids call for the integration of heterogeneous visual styles, which can result in incoherent or conflicting appearances. Lastly, causality and physics demand accurate cause–effect reasoning and physical plausibility, where T2V models typically fail to simulate realistic interactions or transformations. See representative examples in Appendix A.

### 3.2.2 SCENARIO-SPECIFIC REFINEMENT STRATEGIES

In order to facilitate fine-grained prompt refinement strategies for each challenging scenario, we design scenario-specific meta-prompts that guide a rewriting model to generate more precise scene descriptions and richer representations of event dynamics compared with the original prompts, with detailed prompts provided in Appendix E. In addition, we define meta-prompts for non-challenging scenarios to encourage the rewriting model to explicitly specify spatial relationships among objects and temporal dependencies among events.

In abstract descriptions, the rewriting model is guided to translate metaphors and symbolic language into concrete, visually interpretable content while preserving the intended artistic or thematic meaning. For complex spatial relationships, the rewriting model is encouraged to convey precise positions, distances, and orientations among multiple entities to avoid misrepresentations. Multi-element scenes require logically structured descriptions that comprehensively account for all characters, objects, and actions to prevent omissions or disorder. Fine-grained details are emphasized by highlighting subtle visual attributes and incorporating cinematic cues that capture textures, expressions, and other intricate elements. Temporal consistency is maintained by presenting actions in coherent sequences and smooth motion progression, while stylistic hybrids are addressed by harmonizing multiple visual styles, including color palettes, lighting, and composition. Causality and

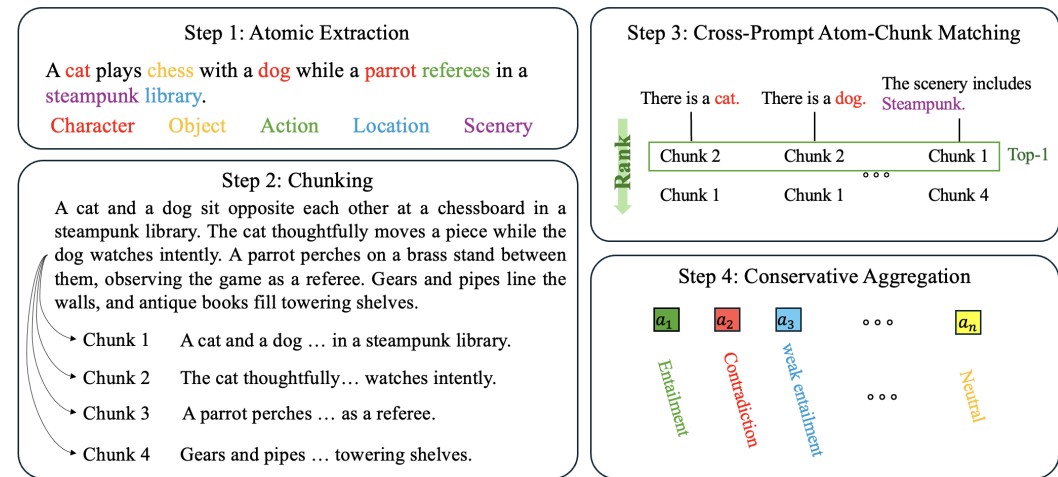

Figure 2: **Pipeline of Refined Prompt Verification via Atomic Extraction and Textual Entailment.** Our goal is to check whether the refined prompt $P_\text{rew}$ faithfully preserves the original intent in $P_\text{in}$. The process consists of four main stages: atomic extraction, semantic chunking, atom–chunk alignment, and conservative aggregation.

physical plausibility are reinforced by explicitly depicting cause–effect relationships, physical interactions, and stepwise morphological changes. For non-challenging scenarios, prompts encourage light refinement to improve clarity while preserving the original intent.

### 3.2.3 SCENARIO CLASSIFICATION

After receiving a user prompt $P_\text{in}$, the scenario classifier $f$ performs an eight-way classification with seven difficult scenarios and a non-difficult class. The classifier classifies the original prompt to the most suitable scenario label $\hat{\mathbf{y}}$ and estimates the confidence of this classification. To ensure the reliability of the scenario classification, we introduce three confidence metrics, i.e., *Maximum Softmax Probability* (MSP), *Margin*, and *Entropy*. Given a set of class probabilities $\{p_i\}_{i=1}^C$ over candidate labels $C$, these metrics can be computed as follows:

$$\text{MSP} = \max_i p_i; \quad \text{Margin} = p_{(1)} - p_{(2)}; \quad \text{Entropy} = -\sum_{i=1}^{C} p_i \log p_i, \quad (1)$$

where $p_{(1)}$ and $p_{(2)}$ denote the top-1 and top-2 probabilities, respectively. Intuitively, a confident prediction is characterized by a high MSP. A large Margin indicates clear separation between the top predictions. A low Entropy reflects a peaked probability distribution. A classification result is considered uncertain if MSP is below a predefined confidence threshold, Margin is small, or Entropy is high.

Once a classification is marked as uncertain, the scenario label is reassigned through a more reliable large language model (e.g., ChatGPT (Achiam et al., 2023), Gemini (Team et al., 2023), or DeepSeek (Liu et al., 2024a)), which performs zero-shot classification based on predefined scenario definitions as a fallback strategy. The final predicted label $\hat{y}$ corresponds to a *scenario-specific meta-prompt*, which is applied to guide prompt refinement. This meta-prompt is then combined with the original user prompt and fed into the prompt refinement model to produce a rewritten prompt $P_\text{rew}$. Then, $P_\text{rew}$ are required to undergo a dedicated prompt verification stage (see Section 3.3). Only validated prompts $P_\text{valid}$ are passed to the downstream T2V generation model, while invalid prompts $P_\text{invalid}$ are discarded. If a rewritten prompt is judged invalid, the refinement model will regenerate an improved prompt according to the verification feedback. For instance, if the rewritten prompt omits semantic information from the original user prompt, an additional instruction reminding the model to preserve this information will be concatenated to the end of meta-prompt before the user prompt.

### 3.3 PROMPT VERIFICATION

In this section, we propose a structured verification framework to rigorously assess whether an expanded T2V generation prompt faithfully retains the semantic essence of its original concise input. Expanded prompts generally incorporate stylistic attributes and narrative embellishments, therefore reliable verification is critical to ensure the quality of T2V generation. As shown in Figure 2, to address this challenge, our method decomposes the process into four well-defined stages: (1) atomic extraction: isolating fundamental entities and actions; (2) semantic alignment: ensuring coverage and consistency of critical elements; (3) dual-model textual entailment: leveraging complementary inference models to evaluate semantic fidelity; and (4) conservative aggregation: synthesizing multi-stage evidence to produce a robust verification judgment.

#### 3.3.1 ATOMIC EXTRACTION FROM ORIGINAL PROMPT

Given an original prompt $P_{\text{user}}$, we define an atomic representation as a minimal semantic tuple:

$$\mathcal{A}(P_{\text{orig}}) = \langle \texttt{characters}, \texttt{objects}, \texttt{actions}, \texttt{locations}, \texttt{scenery} \rangle. \tag{2}$$

Each field in $\mathcal{A}$ is a list that can contain zero, one, or more atomic elements $a_i$, where each element $a_i$ corresponds to a semantically indivisible unit in that category. For example, given the prompt *"A cat plays chess with a dog while a parrot referees in a steampunk library"*, the atomic extractor produces: {characters: [cat, dog, parrot], objects: [chess], actions: [plays, referees], locations: [library], scenery: [steampunk] }. After extracting atomic entities, we expand each atom into a minimal standalone sentence to facilitate similarity computation with the chunked segments of the rewritten prompt. For example, the atom cat is transformed into a simple sentence as "there is a cat". To simplify processing, we discard the dictionary keys and retain the atomic elements themselves, forming a set that is formally represented as $\mathcal{A} = \{a_1, a_2, \cdots, a_n\}$.

#### 3.3.2 REFINED PROMPT CHUNKING

The refined prompt $P_{ref}$ is segmented into non-overlapping chunks $\mathcal{C} = \{c_1, \ldots, c_m\}$. One sentence is treated as an individual chunk unless its length is below a threshold. In that case, it is iteratively merged with the subsequent sentence until the combined length exceeds the threshold.

**The Necessity of Chunking and Atomization.** A naive strategy would be to directly compute the similarity between the embeddings of the original and refined prompts. However, this approach can only provide a coarse measure of semantic closeness and does not clearly determine whether the refined prompt omits critical information from the original. In addition, refined prompts are commonly longer and contain additional stylistic and contextual details, which tend to shift their embeddings away from those of short original prompts even when semantics are faithfully preserved. More importantly, similarity does not distinguish between inclusion and contradiction. For example, the atom "a girl in a red raincoat" and the expansion "a girl in a yellow raincoat running at night" can still exhibit a high embedding similarity, although they are semantically inconsistent.

#### 3.3.3 CROSS-PROMPT ATOM-CHUNK MATCHING

We adopt a stricter entailment-based criterion rather than merely comparing the similarity between the original and refined prompts. Specifically, we evaluate whether the refined prompt entails each atomic constraint. Note that similarity is leveraged only to retrieve candidate text chunks, while the final verification is performed entirely by textual entailment models.

Specifically, given a set of atoms $\{a_i\}_{i=1}^m$ and a set of chunks $\{c_j\}_{j=1}^n$, we embed both atoms and chunks into a shared semantic space using an embedding model $f_{\mathcal{E}}(\cdot)$, and compute pairwise cosine similarities:

$$s_{ij} = \cos\big(f_{\mathcal{E}}(a_i), f_{\mathcal{E}}(c_j)\big).$$

For each atom $a_i$, we typically select the most relevant chunk (i.e., Top-1 retrieval) from the collection:

$$c_i^* = \arg\max_{c \in \{c_j\}} s\big(f_{\mathcal{E}}(a_i), f_{\mathcal{E}}(c)\big), \tag{3}$$

where $c_i^*$ denotes the retrieved chunk with the highest similarity score.

Table 1: Quantitative comparisons on VBench. Boldfaced values indicate the best performance among all methods, while underlined values indicate the second-best performance.

| Method | Average Score | temporal flickering | imaging quality | object class | multiple objects | spatial relationship |
|---|---|---|---|---|---|---|
| Original Prompt | 76.38% | 97.02% | 70.30% | 92.09% | 65.32% | 57.14% |
| Open-Sora | 77.22% | 97.58% | 70.76% | 93.61% | 65.84% | 58.29% |
| RAPO | 79.11% | 97.94% | 71.43% | 97.13% | 66.68% | 62.37% |
| CSAPR | 79.78% | 98.52% | 71.89% | 97.85% | 67.43% | 63.22% |

Table 2: Quantitative comparisons on EvalCrafter. CSAPR consistently achieves better results, demonstrating significant lead on this benchmark.

| Method | Average Score | motion quality | text-video alignment | visual quality | temporal consistency |
|---|---|---|---|---|---|
| Original Prompt | 62.54 | 53.48 | 70.83 | 64.77 | 61.09 |
| Open-Sora | 63.01 | 53.61 | 71.05 | 65.41 | 61.98 |
| RAPO | 63.98 | 53.64 | 74.33 | 65.93 | 62.02 |
| CSAPR | 65.38 | 54.12 | 76.41 | 66.24 | 64.75 |

### 3.3.4 TEXTUAL ENTAILMENT VERIFICATION

To assess whether each atom is semantically preserved by its most relevant retrieved chunk, we employ a *textual entailment verification* module based on Natural Language Inference (NLI). Specifically, for each atom $a_i$ and its top-1 retrieved chunk $c_i^*$, we employ XLM-RoBERTa (Conneau et al., 2020) as the multilingual NLI model to directly predict the semantic relation label from entailment (ENTAIL), neutral (NEUTRAL), or contradiction (CONTRAD).

Verification is evaluated through two primary metrics. Coverage (Cov) quantifies the proportion of atoms that are entailed by their corresponding chunk $\text{Cov} = \frac{|\{a_i : \text{ENTAIL}\}|}{|\mathcal{A}|}$. Contradiction rate measures the proportion of atoms contradicted by their chunk: $\text{Contrad} = \frac{|\{a_i : \text{CONTRAD}\}|}{|\mathcal{A}|}$. Coverage reflects the semantic completeness of the expanded prompt, while contradiction rate highlights conflicts introduced during prompt refinement. When coverage falls below a predefined threshold or contradiction exceeds a threshold, the system triggers the prompt refinement model to regenerate an improved prompt based on the verification feedback. Specifically, if the refined prompt omits semantic content present in the original user prompt, an additional instruction reminding the model to preserve this information will be appended to the end of the meta-prompt preceding the user prompt.

## 4 EXPERIMENTS

In this section, we present the experimental settings with three benchmarks. Then, we demonstrate the main experimental results. Finally, we show an ablation study.

### 4.1 EXPERIMENTAL SETTINGS

**Benchmarks:** We conduct evaluations on VBench (Huang et al., 2024), EvalCrafter (Liu et al., 2024c) and T2V-CompBench (Sun et al., 2025), three state-of-the-art benchmarks to evaluate the quality of T2V generation. VBench provides a systematic evaluation protocol for comprehensive assessment of visual quality, temporal consistency, and content fidelity. EvalCrafter offers a broad suite of metrics to quantify performance across multiple aspects of video generation. T2V-CompBench is a benchmark specifically designed for compositional T2V scenarios.

**Baselines:** We compare three representative prompt refinement approaches in the field of T2V, including: the original prompts, prompt refiner from Open-Sora (Zheng et al., 2024) and Retrieval-Augmented Prompt Optimization (RAPO) (Gao et al., 2025).

**Implementation:** In the prompt refinement setting, we adopt Wan (Wang et al., 2025a) as the T2V backbone for all experiments. For scenario classification, DeBERTa-v3-large (He et al., 2023) is exploited as the primary model, while DeepSeek-V3 (DeepSeek) (Liu et al., 2024a) serves as a zero-shot fallback. Then, we query a large instruction-tuned LLM, i.e., DeepSeek, to generate refined prompts based on scenario-specific meta-prompts (see Appendix E). In the settings of prompt verifi-

Table 3: Quantitative comparisons on T2V-CompBench. CSAPR achieves the highest average score.

| Method | Average Score | consistent attribute | dynamic attribute | action binding | motion binding |
|--------|--------------|---------------------|-------------------|----------------|----------------|
| Original Prompt | 0.412 | 0.628 | 0.254 | 0.478 | 0.290 |
| Open-Sora | 0.433 | 0.672 | 0.269 | 0.493 | 0.298 |
| RAPO | 0.470 | 0.682 | 0.270 | 0.612 | 0.317 |
| CSAPR | **0.501** | **0.724** | **0.289** | **0.641** | **0.352** |

Figure 3: Comparisons of videos generated using Wan (Wang et al., 2025a) conditioned on user-provided prompts and refined prompts from CSAPR.

cation, we employ BGE-M3 (Chen et al., 2024b) as the embedding model for atom-chunk matching in prompt verification. On the textual entailment task, XLM-RoBERTa-large-XNLI (Conneau et al., 2020) serves as the textual entailment model for atom-level entailment checking.

## 4.2 MAIN RESULTS

As shown in Tables 1, 2 and 3, we present a comprehensive quantitative evaluation of our proposed approach, i.e., CSAPR, against baselines across three widely used benchmarks. It can be observed that CSAPR consistently achieves the best overall performance, demonstrating its effectiveness in T2V generation. Specifically, CSAPR attains the highest Average Score of 79.78% on VBench, outperforming RAPO by 0.67%, Open-Sora by 2.56% and the Original Prompt by 3.40%. Similar trends can also be observed on EvalCrafter, where CSAPR achieves the highest Final Sum Score of 260.21, which significantly surpasses other approaches. Notably, it shows marked improvements in text-video alignment and temporal consistency, underscoring its strength in semantic fidelity and dynamic coherence. When it comes to T2V-CompBench focuses on compositional reasoning, CSAPR achieves the best results in consistent attribute binding, dynamic attribute binding, action binding and motion binding, demonstrating strong performance in fine-grained attribute and motion modeling. In particular, CSAPR surpasses the RAPO, Open-Sora, and original prompt by 0.055, 0.092, and 0.113 on T2V-CompBench when it comes to average score.

To provide an intuitive demonstration on the advantages of CSAPR, Figure 3 highlights the effectiveness of CSAPR across four categories of complex scenarios: complex spatial relations, abstract descriptions, multi-entity scenes and temporal consistency. In the case of complex spatial relations, CSAPR achieves two notable improvements over videos generated directly from the user prompt:

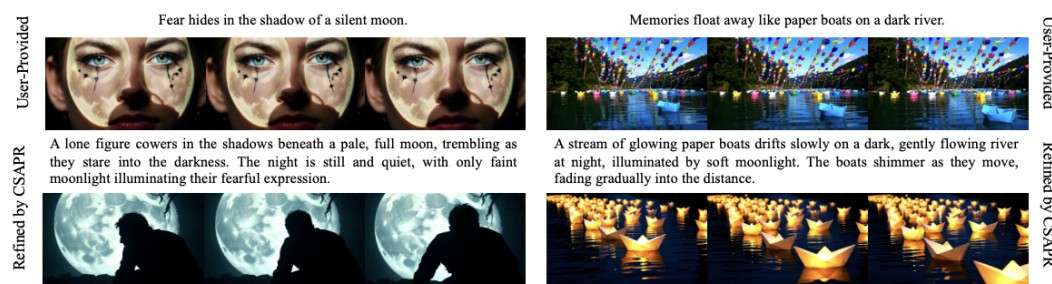

Figure 4: Examples of hallucination elimination after prompt refined by CSAPR.

(1) the parrot is correctly placed at the center (green box) rather than at the edge (red box); and (2) the library is rendered with cyberpunk elements instead of being depicted as a regular library. For abstract descriptions, CSAPR produces richer actions and more vivid visual dynamics. In multi-entity scenes, CSAPR meticulously introduces umbrellas for pedestrians in rainy weather (green box). Finally, under temporal consistency, while the baseline fails to present a fully blossomed flower, CSAPR generates the complete blooming process, resulting in a fully blossomed flower video.

**Hallucination Elimination.** Figure 4 presents examples where simple prompt descriptions lead to hallucinations. In the left panel, the original abstract prompt causes the T2V model to misinterpret the user intent, resulting in an image of a ghostly face illuminated by moonlight. In contrast, the prompt refined by CSAPR provides a concrete depiction aligned with the abstract prompt. Therefore, the T2V model can well understand the desired content and then eliminate hallucinations with the refined prompts of CSAPR. In the right panel, the output based on the original prompt fails to reflect the concept of a dark river, instead producing dense arrays of small colorful flags irrelevant to the prompt. In comparison, the prompt refined by CSAPR conveys a coherent visual narrative that captures the fading of memories and the lingering attachment to the past.

Table 4: Ablation studies of different components in CSAPR on VBench.

| Method | VBench Total Score |
| --- | --- |
| CSAPR | 83.32% |
| Ablate LLM Zero-shot Fallback | 82.45% |
| Ablate Scenario Classification | 81.52% |
| Ablate Prompt Verification | 82.27% |
| Ablate Prompt Refinement | 80.63% |

### 4.3 ABLATION STUDY

To verify the effectiveness of CSAPR, we conduct ablation experiments on the VBench benchmark to examine four key modules in CSAPR. Table 4 presents the ablation results on VBench. The Complete CSAPR achieves the best performance. Either removing the LLM zero-shot fallback or prompt verification leads to minor performance drops (0.87 and 1.05 higher score, respectively). In contrast, discarding the scenario classification (1.80 lower score) or prompt refinement modules (2.69 lower score) results in relatively larger degradations, highlighting their critical roles in CSAPR.

## 5 CONCLUSION

In this paper, we propose a novel prompt refinement approach, i.e., Complex-Scenario-Aware Prompt Refinement (CSAPR), designed for complex T2V generation tasks. CSAPR consists of two key stages: prompt refinement and prompt verification. The prompt refinement stage involves complex scene classification, confidence estimation and LLM-based zero-shot fallback. Following this, the prompt verification stage comprises atomic extraction of the original prompt, new prompt chunking, cross-prompt atom-chunk matching and conservative aggregation to ensure the coherence and completeness. Extensive experiments demonstrate that CSAPR consistently outperforms baseline approaches (up to 1.40% higher in terms of average score).

## REPRODUCIBILITY STATEMENT

To ensure the reproducibility of our work, we have provided comprehensive details about the experimental setup in Section 4, including datasets used, baseline methods, implementation details and evaluation metrics. More details about the implementation can be found in the Appendix. All code, models, and configuration files required to replicate our results are made available in our supplementary materials.

## ETHICS STATEMENT

This paper presents work whose goal is to advance the field of text-to-video generation. The proposed work has the potential to benefit AI agents that possess the function of T2V. Since we employ a training-free prompt refinement model, as long as the T2V generation models (e.g., Wan (Wang et al., 2025a)) and large language models (e.g., DeepSeek (Liu et al., 2024a)) implement appropriate filtering mechanisms for prohibited content, our approach will not lead to the generation of illegal or harmful outputs.

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

# A EXAMPLES FOR COMPLEX SCENARIOS

| Situation | Example |
|---|---|
| Abstract Description | Hope dances in a field of forgotten dreams. |
| Complex Spatial Relationships | A cat plays chess with a dog while a parrot referees in a steampunk library. |
| Multi-Element Scenes | Ten people at a festival, each with different costumes, under fireworks. |
| Fine-Grained Details | A book cover that says Deep Learning 101. |
| Temporal Consistency | A man walking while waving his hand. |
| Stylistic Hybrids | In the style of Van Gogh mixed with cyberpunk neon. |
| Causality and Physics | A glass falling and shattering on the ground. |

Table 5: Challenging scenarios and Example prompts for text-to-video models.

# B ADDITIONAL EXPERIMENTS

## B.1 ADDITIONAL EXAMPLES FOR HALLUCINATION ELIMINATION

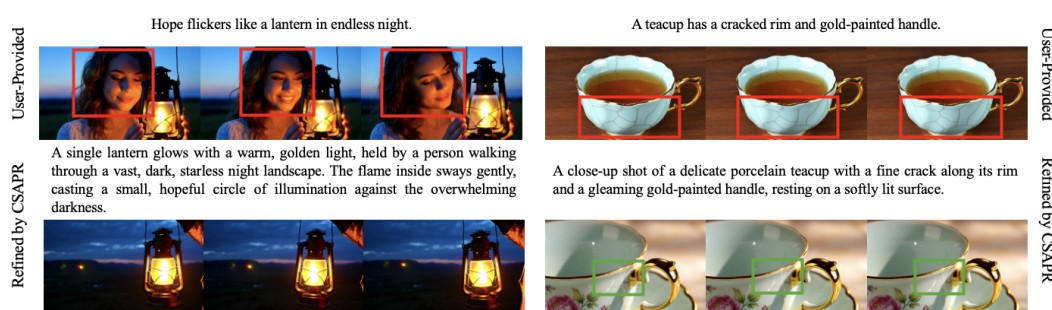

Figure 5: Examples of hallucination elimination after prompt refined by CSAPR. The hallucinated information is highlighted in red, and its elimination or correction is marked in green.

Figure 5 presents examples where simple prompt descriptions lead to hallucinations. In the left panel, although the original prompt does not specify the presence of a human face, the generated video still generates a woman face due to spurious correlations in the training data, where scenes with lamps often occur simultaneously with humans. In contrast, the CSAPR-refined prompt emphasizes scenery, visual context, and atmosphere, resulting in the generated video more aligned with the user intent. In the right panel, the user prompt requests a bowl with crack rim, while the T2V model produces a bowl with grid-like decorations and even tea that was not mentioned. In comparison, the video related to CSAPR-refined prompt yields a bowl with the desired cracks (highlighted in green).

## B.2 ANALYSIS ON SCENARIO DISTRIBUTION AND PROMPT LENGTH

Figure 6 illustrates the scenario distribution in VBench and EvalCrafter across their complete sets of prompts, comprising 946 prompts in VBench and 700 prompts in EvalCrafter. Since neither VBench nor EvalCrafter is specifically designed to evaluate complex scenarios, the majority of prompts in both benchmarks are classified as complex scenarios. Nevertheless, a non-negligible proportion of prompts in both benchmarks are classified into complex scenarios, with a notably higher proportion in EvalCrafter. CSAPR improves the descriptive quality of prompts in complex scenarios.

Figure 7 illustrates the prompt length distributions before and after refinement on VBench and EvalCrafter. For VBench, the original prompts are concentrated within 1–32 tokens, while the refined prompts extend to a broader range of 1–55 tokens. Similarly, the original prompts in EvalCrafter span 1 to 36 tokens, while the refined versions range from 7 to 66 tokens. More importantly, the

distributions exhibit a rightward shift after refinement in both benchmarks, indicating that CSAPR significantly increases prompt length and thereby enriches the information available for video content description.

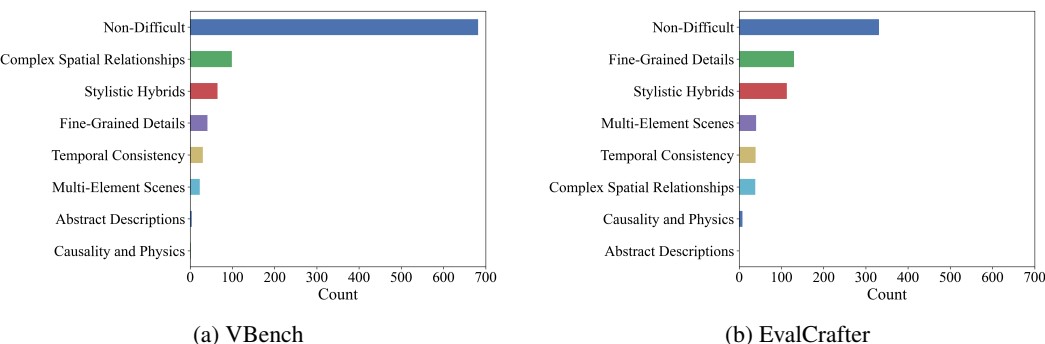

(a) VBench

(b) EvalCrafter

Figure 6: Statistics of scenario classification results.

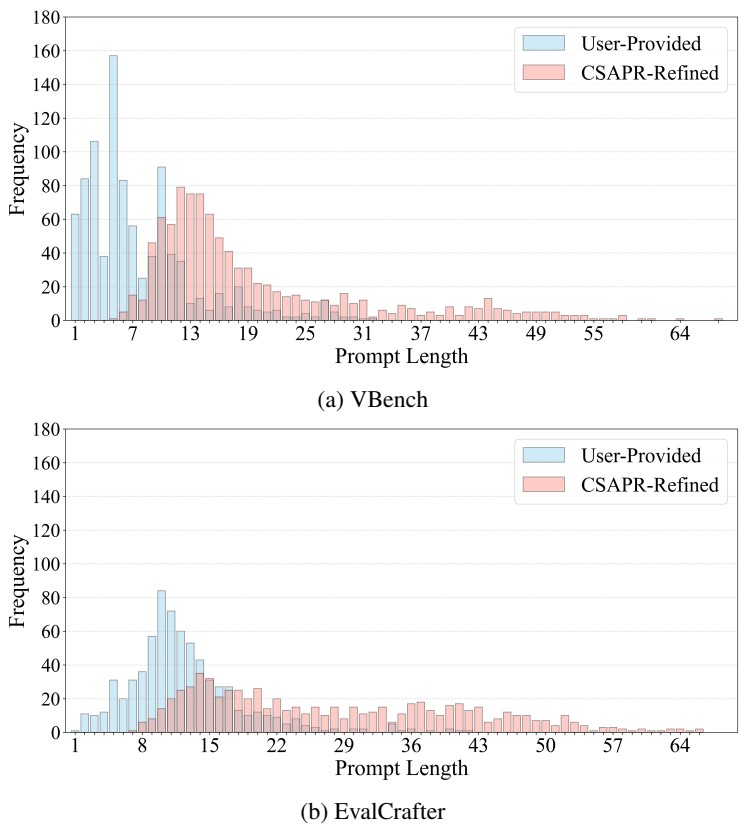

(a) VBench

(b) EvalCrafter

Figure 7: Statistics of prompt length measured in words.

## C  HYPERPARAMETER SETTINGS

In this work, we employ MSP, Margin and Entropy to measure the confidence of scenario classification (see Section 3.2.3), and their thresholds are set to 0.6, 0.2, and 1.5, respectively. The entailment threshold and contradiction threshold are set to 0.7 and 0.3, respectively.

# D  INSTRUCTION FOR SCENARIO CLASSIFICATION

---

**Prompt for Scenario Classification**

```
You are a few-shot classifier for Text-to-Video (T2V) prompt *
    difficulty scenarios*.
Return ONLY a valid JSON object of the exact form:
{"label": "<one of SCENARIO_LABELS>", "reason": "<short phrase (<
    = 20 words)>"}

Allowed labels (must match EXACTLY one string in SCENARIO_LABELS)
    :
1) Abstract Descriptions
2) Complex Spatial Relationships
3) Multi-Element Scenes
4) Fine-Grained Details
5) Temporal Consistency
6) Stylistic Hybrids
7) Causality and Physics
8) non-difficult

## Task
Given a short English prompt P_in, decide which single label best
     describes the dominant difficulty that a T2V model would face
     when generating a video.

## Diagnostic definitions:
- Abstract Descriptions: Figurative language, metaphors, emotions
     as objects, surreal imagery.
- Complex Spatial Relationships: Explicit positions/orientations
    between >=2 entities; lots of prepositions ("on top", "behind"
    , "between").
- Multi-Element Scenes: >=3 different entities or activities;
    dense environments with many elements in one shot.
- Fine-Grained Details: Micro-level attributes (textures, tiny
    objects, reflections, accessories); often close-up.
- Temporal Consistency: Clear time progression or motion over
    time (bloom, melting, time-lapse).
- Stylistic Hybrids: Mixing multiple visual or artistic styles;
    style blending is central.
- Causality and Physics: Cause-effect chains or physical forces (
    gravity, splashes, collisions).
- non-difficult: None of the above applies.

## Tie-breaking rules:
1) Figurative language dominates -> Abstract Descriptions
2) Spatial focus dominates -> Complex Spatial Relationships
3) Many varied elements, no strong spatial focus -> Multi-Element
     Scenes
4) Close-up or micro details dominate -> Fine-Grained Details
5) Time progression dominates -> Temporal Consistency
6) Mixed styles dominate -> Stylistic Hybrids
7) Physics/cause-effect dominate -> Causality and Physics
8) Otherwise choose non-difficult.
```

```
## Few-shot examples (prompt -> label):
- "Hope dances in a field of forgotten dreams." -> Abstract
    Descriptions
- "A cat and a dog sit back-to-back; a parrot hovers above." ->
    Complex Spatial Relationships
- "A neon street with vendors, robots, and flashing billboards."
    -> Multi-Element Scenes
- "A gold pocket watch with a cracked rim on velvet." -> Fine-
    Grained Details
- "A bud opens into a flower in slow motion." -> Temporal
    Consistency
- "A medieval castle with neon cyberpunk signs." -> Stylistic
    Hybrids
- "A glass tips; wine splashes and forms ripples." -> Causality
    and Physics
- "A child runs across a field." -> non-difficult

Classify this prompt:
P_in: {P_in}
```

# E   META PROMPTS

**Meta Prompt for Abstract Description**

You are a prompt refinement expert for text-to-video generation. You are given a user-provided prompt that contains **abstract or metaphorical descriptions**. Your task is to rewrite and optimize this prompt for a text-to-video generation model.
Follow these requirements:

1. **Clarify abstract imagery:** Translate metaphors, symbolism, or abstract phrases into literal visual elements (characters, objects, actions, settings).

2. **Be explicit and detailed:** Specify scene components clearly.

3. **Keep cinematic focus:** Include camera framing, lighting, or style cues only if they are implied by the original prompt.

4. **Maintain artistic tone:** Keep the emotional or thematic essence of the metaphor while improving visual clarity.

5. **Limit length:** The rewritten prompt must be concise, under 100 words, and multiple sentences are allowed.

6. **No extra interpretation:** Do not explain, comment, or add content. Only output the rewritten prompt.

Only output a single, polished rewritten prompt that meets all requirements.

---

**Meta Prompt for Complex Spatial Relationships**

You are a prompt refinement expert for text-to-video generation. You are given a user-provided prompt that contains **complex spatial relationships** between objects, characters, and environments. Your task is to rewrite and optimize this prompt for a text-to-video generation model.
Follow these requirements:

1. **Emphasize spatial clarity:** Explicitly describe positions, distances, and relative orientations of elements in the scene.

2. **Position characters by relationship:** Place adversarial characters on opposite sides. Place non-adversarial characters between the adversarial characters.

3. **Assign appropriate actions:** Define suitable and clear movements or actions for each character.

4. **Simplify sentence structure:** Use short sentences or clear clauses to avoid ambiguity.

5. **Maintain key details:** Preserve all essential objects, actions, characters, and environments.

6. **No Extra Interpretation:** Do not explain, comment, or add content. Only output the rewritten prompt.

7. **Limit length:** The rewritten prompt must be **concise, under 100 words**, and multiple sentences are allowed.

Only output a single polished rewritten prompt that meets all requirements.

---

**Meta Prompt for Multi-Element Scenes**

You are a prompt refinement expert for text-to-video generation. You are given a user-provided prompt that describes **multi-element scenes** with multiple characters, objects, actions, and locations. Your task is to rewrite and optimize this prompt for a text-to-video generation model.
Follow these requirements:

1. **Multiple sentences allowed:** Use concise sentences or separated clauses to describe scenes clearly.

2. **Preserve all key elements:** Keep essential characters, objects, settings, and relationships.

3. **Simplify structure:** Avoid unnecessary adjectives or complex phrasing.

4. **Ensure temporal and spatial clarity:** Present events in a logical and visually coherent order.

5. **No Extra Interpretation:** Do not explain, comment, or add content. Only output the rewritten prompt.

6. **Limit length:** The rewritten prompt must be **concise, under 100 words**, and multiple sentences are allowed.

Only output a single polished rewritten prompt that meets all requirements.

**Meta Prompt for Fine-Grained Details**

You are a prompt refinement expert for text-to-video generation. You are given a user-provided prompt that contains descriptions of a **Scene with Fine-Grained Details**. Your task is to rewrite and optimize this prompt for a text-to-video generation model.
Follow these requirements:

1. **Preserve Fine-Grained Details:** Keep all essential visual attributes (colors, textures, facial expressions, clothing, environmental elements, etc.) while removing irrelevant or repetitive details.

2. **Enhance Visual Clarity:** Use precise and descriptive language to clearly define characters, objects, actions, and spatial relationships, making the scene easy for the model to interpret.

3. **Add Cinematic Guidance:** Optionally introduce cinematic elements like lighting, camera movement, focus depth, or shot composition to improve video realism.

4. **Maintain Logical Structure:** Ensure actions and events are described in chronological order with clear transitions, avoiding ambiguity or contradictions.

5. **Optimize for Video Generation:** Emphasize motion cues, scene continuity, and environmental context so the model can generate smooth, coherent multi-frame sequences.

6. **No Extra Interpretation:** Do not explain, comment, or add content. Only output the rewritten prompt.

7. **Limit length:** The rewritten prompt must be **concise, under 100 words**, and multiple sentences are allowed.

Only output a single polished rewritten prompt that meets all requirements.

**Meta Prompt for Temporal Consistency**

You are a prompt refinement expert for text-to-video generation. You are given a user-provided prompt that requires **temporal consistency**, meaning the scene involves actions, events, or changes that must follow a logical and coherent timeline across frames. Your task is to rewrite and optimize this prompt for a text-to-video generation model.
Follow these requirements:

1. **Be Clear and Explicit:** Turn ambiguous or compressed descriptions into precise phrases.

2. **Be Scene-Oriented:** Clearly separate and describe characters, objects, locations, and actions.

3. **Follow Logical Order:** Present elements in a clear sequence (foreground → background; primary → secondary; chronological actions).

4. **Preserve All Key Details:** Keep every important visual detail while removing redundancies.

5. **Include Style and Lighting:** Explicitly state any implied visual style, palette, or lighting.

6. **No Extra Interpretation:** Do not explain, comment, or add content. Only output the rewritten prompt.

7. **Limit length:** The rewritten prompt must be **concise, under 100 words**, and multiple sentences are allowed.

Only output a single polished rewritten prompt that meets all requirements.

**Meta Prompt for Stylistic Hybrids**

You are a prompt refinement expert for text-to-video generation. You are given a user-provided prompt that contains **Stylistic Hybrids**, meaning multiple artistic or visual styles combined in one scene. Your task is to rewrite and optimize this prompt for a text-to-video generation model.
Follow these requirements:

1. **Style Clarity:** Clearly describe each style and how they interact.

2. **Scene Composition:** Specify key subjects, actions, and environments in short, direct phrases.

3. **Visual Consistency:** Resolve ambiguity about style blending or scene layout.

4. **Compactness:** Use minimal yet descriptive language; no filler words.

5. **Model-Friendly Syntax:** Output a single well-structured description in multiple concise sentences.

6. **No Extra Interpretation:** Do not explain, comment, or add content. Only output the rewritten prompt.

7. **Limit length:** The rewritten prompt must be **concise, under 100 words**, and multiple sentences are allowed.

Only output a single polished rewritten prompt that meets all requirements.

---

**Meta Prompt for Causality and Physics**

You are a prompt refinement expert for text-to-video generation. You are given a user-provided prompt that contains **Causality and Physics** elements (e.g., cause-effect relationships, realistic object interactions, motion, forces). Your task is to rewrite and optimize this prompt for a text-to-video generation model.
Follow these requirements:

1. **Preserve Meaning:** Retain all key entities, actions, and causal relationships.

2. **Physics Clarity:** Clearly state motion, timing, and forces.

3. **Morphological Changes:** Emphasize transformations in object shape, size, or state over time.

4. **Logical Flow:** Present actions in chronological order.

5. **No Extra Interpretation:** Do not explain, comment, or add content. Only output the rewritten prompt.

6. **Limit length:** The rewritten prompt must be **concise, under 100 words**, and multiple sentences are allowed.

Only output a single polished rewritten prompt that meets all requirements.

---

**Meta Prompt for Non-difficult Scenario**

You are a prompt refinement expert for text-to-video generation. You are given a user-provided prompt that is **simple and straightforward**, without abstract concepts, complex spatial reasoning, or other difficult elements. Your task is to **lightly refine and optimize** this prompt for a text-to-video generation model.
Follow these requirements:

1. **Preserve Original Intent:** Keep all entities, actions, and scene elements exactly as described, without adding or removing content.

2. **Improve Clarity:** Rewrite in clear, simple language to eliminate ambiguity or vagueness.

3. **Model-Friendly Syntax:** Ensure the prompt is straightforward for machine interpretation and avoid figurative language or unnecessary modifiers.

4. **Direct Scene Description:** Describe the scene plainly, focusing only on necessary visual elements.

5. **No Extra Interpretation:** Do not explain, comment, or add content. Only output the rewritten prompt.

6. **Limit length:** The rewritten prompt must be **concise, under 80 words**, and multiple sentences are allowed.

Only output a single polished rewritten prompt that meets all requirements.

## F  THE USE OF LARGE LANGUAGE MODELS

In this work, LLMs serve as the backbone for scenario classification and prompt rewriting, and are also employed to generate instructions for scenario classification (see Appendix D). Outside of these uses, LLMs are not employed in a centralized manner.

