# OpenReview forum: "CSAPR: Complex-Scenario-Aware Prompt Refinement for Text-to-Video Generation"
_ICLR.cc/2026/Conference — ICLR 2026 Conference Withdrawn Submission_

### Official Review · Reviewer_LCdw · 2025-10-25

**Soundness:** 2
**Presentation:** 2
**Contribution:** 2
**Rating:** 2
**Confidence:** 4

**Summary:**

This paper proposes a prompt refinement approach, CSAPR for complex T2V generation. CSAPR consists of prompt refinement and prompt verification: prompt refinement classifies user prompts  into representative categories and applies rewriting strategies; prompt verification extract semantic atoms with decomposed chunks to preserve semantics while keeping consistency. Experiments show improvements over three benchmarks.

**Strengths:**

1. The paper is well-written and easy-to-follow.
2. The experiments are though and extensive.

**Weaknesses:**

1. The idea of prompt refinement/verification is not novel, which is often applied in T2I and T2V generation [A, B]. The claims of "the first work that investigate prompt refinement" is not proper.
2. The method seems to be engineering (combined step by step) than novel.
3. For cases that video generation models cannot generate well even with perfect prompts, the method fails.

[A] Ji Y, Zhang J, Wu J, et al. Prompt-a-video: Prompt your video diffusion model via preference-aligned llm[C]//Proceedings of the IEEE/CVF International Conference on Computer Vision. 2025: 18725-18735.
[B] Tiehan Fan, Kepan Nan, et al. Instancecap: Improving text-to-video generation via instanceaware structured caption.

**Questions:**

1. Why classify prompts into eight representative categories? How to divide the eight categories, and is it proper and comprehensive to cover all situations?

2. How to extract atomic parts? How to ensure accuracy?

3. How many sentences are iteratively merged in average in refined prompt chunking?

---

### Official Review · Reviewer_kZGb · 2025-10-30

**Soundness:** 2
**Presentation:** 2
**Contribution:** 2
**Rating:** 2
**Confidence:** 4

**Summary:**

This paper introduces CSAPR, a two-stage framework to improve Text-to-Video (T2V) generation for complex prompts. First, the "Prompt Refinement" stage classifies a prompt into one of eight categories (including seven complex scenarios like temporal consistency or causality)  and uses a specific meta-prompt to guide an LLM in rewriting it. Second, the "Prompt Verification" stage ensures the new prompt remains faithful to the original by extracting and aligning "semantic atoms" and "chunks" . Experiments on three benchmarks (VBench, EvalCrafter, T2V-CompBench) showed that CSAPR consistently outperformed baselines, achieving up to a 1.40% improvement in average score.

**Strengths:**

This paper introduces a two-stage prompt refinement strategy for the text-to-video (T2V) task. The authors summarize seven challenging scenarios in the T2V task and propose a prompt refinement strategy and a verification strategy to solve them. Experiments on Vbench and EvalCrafter demonstrate its effectiveness.

**Weaknesses:**

1. Lack of Novelty: The framework is an engineering combination of existing, mature techniques (e.g., classifiers, LLM-based prompt rewriting, and NLI for verification) rather than a novel scientific contribution.
2. Reliance on a Fixed, Manual Scenario List: The framework's effectiveness is dependent on a manually-curated list of "7 complex scenarios" . This design is rigid and incomplete, leaving it unable to adapt or handle new types of complexity (e.g., logical reasoning, counting) not on its list.
3. Limited Experimental Validation: The experiments were conducted solely on the "Wan" T2V model. This lack of validation on SOTA models (e.g., Wan2.2, Sora) raises concerns about the method's generalizability.

**Questions:**

see weakness above

---

### Official Review · Reviewer_ATjf · 2025-11-01

**Soundness:** 2
**Presentation:** 3
**Contribution:** 2
**Rating:** 4
**Confidence:** 3

**Summary:**

The paper proposes CSAPR, a two-stage Complex-Scenario-Aware Prompt Refinement framework for text-to-video (T2V). The refinement stage first classifies a user prompt into one of several difficult scenario types, then rewrites the prompt using a scenario-specific meta-prompt. The verification stage checks that the rewritten prompt preserves the original semantics. The validated refined prompt is then sent to a T2V model. Experiments on VBench, EvalCrafter, and T2V-CompBench show consistent gains over baseline refiners.

**Strengths:**

+ Targeted refinement with a safety check: The framework uses the scenario-specific meta-prompts and, based on the classifier’s confidence, falls back to an LLM when uncertain. This is practical and likely improves recall of hard cases.

+ Ablations: Removing scenario classification or refinement hurts more than removing fallback or verification, supporting the design choices.

**Weaknesses:**

- Benchmarks are standard, but many prompts are not explicitly designed for complex scenarios. Scenario statistics (Figure 6) also show that many are classified as non-difficult scenarios. It would be good to add a curated set of real user prompts for the seven categories and report per-category performance.

- The uncertainty is measured by predefined thresholds. How robust and stable is this approach across prompts and seeds? Please provide an error analysis (confusion and failure cases) and a sensitivity study of these thresholds.

- Cost: CSAPR is based on a classifier, LLM fallback, rewriting, embeddings, and NLI. Please report end-to-end latency, a per-component speed, and throughput compared to the baselines.


- Generality: The method relies on the T2V backbone (Wan), scenario classifier (DeBERTa-v3-large), LLM (DeepSeek-V3) for fallback and the prompt generation, prompt verifier (BGE-M3), and NLI (XLM-RoBERTa-large-XNLI). How sensitive are the results if these components are replaced with other open models?

**Questions:**

See weaknesses.

---

### Official Review · Reviewer_pY9v · 2025-11-02

**Soundness:** 2
**Presentation:** 2
**Contribution:** 1
**Rating:** 2
**Confidence:** 4

**Summary:**

The paper proposes a novel two-stage framework named Complex-Scenario-Aware Prompt Refinement (CSAPR) to enhance user-provided prompts specifically for Text-to-Video (T2V) generation models. The system first uses a Large Language Model (LLM) to classify the user prompt into one of eight categories, including seven pre-defined challenging scenarios (e.g., Temporal Consistency, Causality). Based on this classification, a targeted meta-prompt guides the LLM to rewrite and refine the original prompt. The second stage, Prompt Verification, uses Natural Language Inference (NLI) to ensure the refined prompt retains the original semantic content and does not introduce contradictions, instructing the LLM to regenerate if verification fails. Experiments show CSAPR consistently improves T2V generation quality in complex scenarios across three benchmarks.

**Strengths:**

1. Targeted T2V Focus: It successfully identifies the prompt-level difficulties specific to T2V, such as temporal coherence and causality, which are often overlooked by T2I refinement methods.
2. Valuable Taxonomy: The paper provides a clear taxonomy of seven challenging T2V prompt scenarios, which is a useful contribution for researchers benchmarking the controllability of future T2V models.

**Weaknesses:**

1. Lack of Novelty in Method: The core technique is an application of well-established components (LLM-based rewriting and NLI-based verification). The novelty is mainly in the composition and the manual definition of T2V challenge categories, not in a novel algorithmic breakthrough.
2. Does Not Address Fundamental T2V Limitation: Refinement only addresses the input-side (prompt). The fundamental failures in T2V generation (e.g., severe object deformation, scene flickering, or physics violation) are internal limitations of the diffusion model architecture itself, which prompt engineering cannot fundamentally solve.
3. Limited Quantitative Improvement: While the improvements are consistent, despite the complex setup, the reported gains are small (e.g., up to 1.40% in average score in EvalCrafter).
4. Reliance on Black-Box LLM and Meta-Prompts: The core prompt transformation relies on a large, instruction-tuned LLM (DeepSeek) guided by manually designed meta-prompts. This reliance makes the system sensitive to the specific LLM/model family used and the human engineering effort, and the paper lacks a detailed sensitivity analysis on this component.

**Questions:**

1. The system's effectiveness relies heavily on a specific LLM and manually engineered meta-prompts. How sensitive is CSAPR's performance to different LLM models (e.g., GPT-4 vs. Llama 3) or minor variations in the meta-prompt structure? Can the authors provide a robustness study for this key component?

---

### Note · Authors · 2025-11-23

I have read and agree with the venue's withdrawal policy on behalf of myself and my co-authors.